# Electrochemical Performance Enhancement of Micro-Sized Porous Si by Integrating with Nano-Sn and Carbonaceous Materials

**DOI:** 10.3390/ma14040920

**Published:** 2021-02-15

**Authors:** Tiantian Yang, Hangjun Ying, Shunlong Zhang, Jianli Wang, Zhao Zhang, Wei-Qiang Han

**Affiliations:** School of Materials Science and Engineering, Zhejiang University, Hangzhou 310027, China; dayday_y@foxmail.com (T.Y.); zhangshunlong@zju.edu.cn (S.Z.); 11726038@zju.edu.cn (J.W.); 11826056@zju.edu.cn (Z.Z.)

**Keywords:** Si/Sn@G-C composite, Lithium-ion battery, tin, porous silicon, anode material

## Abstract

Silicon is investigated as one of the most prospective anode materials for next generation lithium ion batteries due to its superior theoretical capacity (3580 mAh g^−1^), but its commercial application is hindered by its inferior dynamic property and poor cyclic performance. Herein, we presented a facile method for preparing silicon/tin@graphite-amorphous carbon (Si/Sn@G–C) composite through hydrolyzing of SnCl_2_ on etched Fe–Si alloys, followed by ball milling mixture and carbon pyrolysis reduction processes. Structural characterization indicates that the nano-Sn decorated porous Si particles are coated by graphite and amorphous carbon. The addition of nano-Sn and carbonaceous materials can effectively improve the dynamic performance and the structure stability of the composite. As a result, it exhibits an initial columbic efficiency of 79% and a stable specific capacity of 825.5 mAh g^−1^ after 300 cycles at a current density of 1 A g^−1^. Besides, the Si/Sn@G–C composite exerts enhanced rate performance with 445 mAh g^−1^ retention at 5 A g^−1^. This work provides an approach to improve the electrochemical performance of Si anode materials through reasonable compositing with elements from the same family.

## 1. Introduction

Lithium-ion batteries (LIBs) affect many aspects of our lives because of their high energy density, low self-discharge rate and good rate capacity [1,2,3]. However, the development of related equipment puts forward higher requirements for the performance of LIBs [4,5,6,7,8,9,10]. Graphite has been used in LIBs as anode material for decades because of its stable capacity retention, but its theoretical capacity of 372 mAh g^−1^ limits the further development of LIBs with higher energy density and security [11,12,13,14].Therefore, much efforts have been devoted to develop new anode materials, such as Si [15], Sn [16] and metal oxides [17,18,19,20,21,22,23]. Among all the candidates, silicon with high specific capacity (3580 mAh g^−1^) [24], moderate operation potential (around 0.4 V vs. Li/Li^+^), environmentally benign nature and abundant reserves is considered as one of the most promising next generation anode materials [25,26]. Despite these advantages, some defects prevent silicon anode from being widely used: its low electronic conductivity (2 × 10^3^ Ω·m) and structural instability upon lithiation result in inferior rate performance and rapid capacity fading [27].

In order to alleviate the huge change in the volume during cycles and increase the electrical conductivity of Si, several approaches have been discussed, including constructing different structures of silicon and designing Si-based composites [28]. Various silicon-based nanostructures have been proposed, such as nanoparticles [29,30,31], nanowires [32,33,34], porous structures [35,36,37], yolk-shell constructions [38]. These structures exhibit good structural stability, fast ion diffusion path and led to ideal electrochemical performance, but most of them lack potential of commercial application because of the high cost of the methodology or the inability to large-scale preparation [39]. The micro-sized porous Si with high performance has exhibited hopeful prospect of practical application and scalable production [12].

Reasonable compositing modification is also an important way to promote the electrochemical behavior of Si anode materials. [40,41,42,43]. For example, Yang [16] synthesized a Si/Sn@C-G composite by the method of high energy ball milling and annealing, the composite shows stable reversible capacity of 612.6 mAh g^−1^ at a current density of 1 A g^−1^ and a high initial columbic efficiency of 81.5%. Zhong et al. [44] used nano-sized silicon particles, tin dichloride and PVP to synthesize a Si/Sn@C anode that has good electrochemical performance. Yang et al. [39] prepared a Si/C-G anode by the way of ball milling and carbon pyrolyzing, the composite has a capacity of 445 mAh g^−1^ at a current density of 0.5 A g^−1^ with a high capacity retention after 200 cycles. In addition, the way of combining Si with other active or inactive metals (such as Sn [35], Co [45,46], Ag [47,48,49], Ni [50], Cu [51,52,53], et. al.) has also been widely reported for LIBs. Hao et al. [35] fabricated hierarchical macroporous Si/Sn composites via facile and green dealloying process by etching SiSnAl alloy precursors, the composite delivers a capacity of 1600 mAh g^−1^ at 200 mA g^−1^ in the period of 70 cycles. Zhao et al. [54] synthesized an Ag–Si core–shell nanowall arrays by displacement reaction and subsequent RF-sputtering deposition, the composite shows a specific capacity of 1500 mAh g^−1^ at 2100 mA g^−1^. Li et al. [55] fabricated a Si-Cu composite by etching and electroless plating, the composite presents a specific capacity of 1651 mAh g^−1^ after 150 cycles with a columbic efficiency of 99%. Although many attainments have been achieved in the development of Si anode materials, a facial and low-cost modification way is still desired [56,57]. The metallic Sn in the same group has similar lithiation mechanism and electrochemical potentials with that of Si, and shows much better electro conductibility [58,59]. Hence, Sn can work as an efficient agent to improve the kinetic properties of Si. Herein, we design and synthesize silicon/tin@graphite-amorphous carbon (Si/Sn@G-C) composites and assesse their performance as anode for LIBs. The reasonably designed ternary composites exhibit enhanced performance than the contrast samples, which may result from the kinetic improvement and structural reinforcement of nano-Sn and composited carbon.

## 2. Materials and Methods

### 2.1. Material Preparation

Micro-sized porous silicon: the silicon particles are prepared by the way of etching Fe-Si alloy, similar to our previous report [60]. The obtained micro-sized porous silicon presents a unique hierarchical pore structure with coexisting micro-meso-macropores. The specific surface area of as-synthesized micro-sized porous silicon is 17.1 m^2^ g^−1^ (calculated from BET data).

Si/Sn@G–C, Si@G–C and Si/Sn@G composite: micro-sized porous silicon, artificial graphite (>99.5%, Xianding Biological Technology Co., Ltd., Shanghai, China), stannic chloride (99%, Alfa Aesar, Haverhill, MA, USA), polyvinyl alcohol (molar weight 31,000, Aladdin, China), are used as raw materials for the synthesis of the Si/Sn@G-C composites. Typically, 0.4 g micro-sized porous silicon and 0.1 g SnCl_2_·2H_2_O are weighted and added into 20 mL distilled H_2_O and magnetic stirred for 60 min at room temperature. The treated silicon particles are then rinsed with distilled H_2_O and treated through filtering. The particles obtained are further ball milled with graphite and the polyvinyl alcohol with a weight ratio of 4:4:2 at 400 rpm for 5 h in a ball mill machine (PM200, Retsch GmbH Inc., Haan, Germany), the weight ratio of milling balls to mixture is 20:1. Finally, the mixture above is pyrolyzed in a tube furnace (BTF-1200C, BEQ Equipment Technology Co., Ltd., Anhui, China) at 750 °C for 3 h (Ar, 100 mL min^−1^) to obtain the Si/Sn@G-C products. The composite treated by the same processes without SnCl_2_ is named Si@G-C and the composite prepared without polyvinyl alcohol is called Si/Sn@G.

### 2.2. Material Preparation

The morphology of the composites is observed by the scanning electron microscopy (SU-8100, Hitachi Manufacturing Co., Ltd., Tokyo, Japan), equipped with an energy dispersive spectroscopy system. The X-ray diffractometer (XRD-6000, Shimadzu Co., Ltd., Tokyo, Japan) with a Cu Kα radiation source (radiation = 0.154 nm, 2θ = 10°~90°) and the Raman spectrum are used to analyze the composition of the as prepared materials, which is operated on a Renish in Via Raman system and the Raman shifts are collected in the range of 1000~2000 cm^−1^. The pore size distribution and the surface area of the composites is evaluated by the analysis of nitrogen adsorption-desorption data on an ASAP 2020M (Micromeritics Instrument Corp., Norcross, GA, USA). Thermogravimetric analysis (TGA) was carried out using Perkin Elmer Diamond TG/DTA (Perkin Elmer Co., Ltd, Waltham, Massachusetts, USA) instrument at a heating rate of 10 °C min^−1^ from 20 °C to 900 °C under air.

### 2.3. Electrochemistry Measurements

To measure the electrochemical performance of the obtained sample, an aqueous slurry contains the active materials, carboxyl methyl cellulose (Aldrich, Shanghai, China) and carbon black (Alfa Aesar, Haverhill, MA, USA) with the weight ratio of 8:1:1 is prepared. The slurry is stirred for over 6 h and uniformly casted onto a thin copper foil (40 mm × 100 mm) and dried in vacuum at 70 °C for 12 h. Then the film is punched into circular discs for the future fabrication. The coin-type cells are assembled with lithium foil as the counter electrode, a poly-propylene film (Celgard-2400) as the separator and 1 M LiPF_6_ solution in a mixture of dimethyl carbonate(DMC), fluoroethylene carbonate (FEC), and ethyl methyl carbonate(EMC) as the electrolyte (DMC/FEC/EMC, 1:1:1 in volume, Shanshan Battery Material Co., Ltd., Dongguan, China). The coin cells are assembled in an argon-filled glove box (Mikrouna, China). The cells are charged and discharged on a LAND instrument (Lanhe-CT3100A, Wuhan Landian Electronics Co., Ltd., China) within the voltage range of 0.1~1.2V. The cyclic voltammogram (CV) is carried out at a scan rate of 0.1 mV s^−1^ within a potential range of 0.01~1.5 V vs. Li/Li^+^. The EIS measurement is carried out at open circuit potential over the frequency range from 0.01 Hz to 100 kHz with an amplitude of 5 mV. The CV and EIS are measured by an electrochemical workstation (Solartron-1470E, AMETEK, Inc., Champaign, IL, USA).

## 3. Result and Discussion

Figure 1 shows the preparation scheme, XRD patterns and Raman spectra of the sample, respectively. As shown in Figure 1b and Appendix A, as-prepared porous Si and Si/Sn@G-C exhibits a series of diffraction peaks, in which the relative strong peaks around 2θ = 28.4°, 47.3°, 56.1°, 73.3°, 88.0° can be assigned to the (111), (220), (311), (331) and (422) planes of Si (PDF#27-1402), while diffraction peaks at 2θ = 30.6°, 32.0°, 43.9°, 44.9°, 55.3°, 62.5°, 63.8°, 64.6° and 72.4° in Si/Sn@G-C are corresponded to the (200), (101), (220), (211), (301), (112), (400), (321) and (420) planes of Sn, suggesting the successful decoration of nano-Sn on porous Si. Several peaks at 2θ = 26.4°, 44.4° belong to the (002) and (101) planes of the graphite (PDF#41-1487) (Appendix A). According to the TGA test, whose result is presented in Appendix A, the content of carbon in Si/Sn@G-C composite is 53.2 wt.%. The XRD results of other composites (Si/Sn, Si@G-C and Si/Sn@G) are shown in Appendix A All these evidences suggest the Si/Sn@G-C composite is successfully prepared.

The Raman spectra are recorded in Figure 1c, all the composites have two peaks (ca. 1342 cm^−1^ and 1589 cm^−1^), which are the D and G bands of carbon materials, respectively. The D-band is related to the disordered structure of carbon and always can be found in amorphous or disorder carbon, while G-band reflects the bond stretching of the sp^2^ carbon atoms [61]. The ratio of I_D_ to I_G_ (I_D_:I_G_) indicates the graphitization extent of materials, a low I_D_:I_G_ implies a high extent of graphitization. With the addition of amorphous carbon, the I_D_:I_G_ ratio increases, which means the decrease of graphitization extent of Si/Sn@G-C in comparison to Si/Sn@G. Besides, the result shows that the presence of Sn has little effect on the graphitization degree.

We use the scanning electron microscopy (SEM) to collect the morphology information of materials, the top-down SEM images of the composites and the EDS result of Si/Sn@G-C composite are shown in Figure 2, and the element content obtained by EDS is presented in Appendix A. Figure 2a and Appendix A reveal the SEM images of the micro-sized porous Si and the graphite. As shown in Figure 2a, the silicon shows particle size with micro-scale, without obvious agglomeration. It is indicated in Appendix A, the particle size of artificial graphite is around 10 μm. Figure 2b–e and Appendix A are the SEM images of Si@G-C, Si/Sn@G and Si/Sn@G-C composites, respectively. Figure 2d,e show typical SEM images of Si/Sn@G-C composite, it is observed that the graphite undergoes fierce breakage during ball milling. According to the element mapping in Figure 2f–i, the Si, Sn and C elements distribute uniformly in the composite, suggesting the components of Si/Sn@G-C composite are evenly mixed. The SEM image of the Si@G-C composite (Figure 2b) is similar to that of Si/Sn@G-C because the hydrolyzed Sn particles are small in size and attached to the surface of Si and cannot be easily found in SEM image [62]. The existence of nano-Sn has been proved by the XRD and SEM-EDS. As shown in Figure 2c, the amorphous carbon can effectively encapsulate the Si/Sn nanoparticles within it. It is reported that the addition of amorphous carbon on the surface of the composite will prevent the silicon particles from separation by providing effective constrain force and is beneficial for the electronic transmission, so it can improve the electrochemical performance of the material [63].

The transmission electron microscope (TEM) is used to get more detailed information about the morphology and structure of the composite, and the result is given in Figure 3. As shown in Figure 3a, the Si/Sn@G-C is composed of carbon outerwear and grain kernel. Figure 3b–e show the HRTEM images and the selected area electron diffraction (SAED) pattern of the composite, from which we can clearly distinguish the different components of the composite. The amorphous carbon is around 6–10 nm in thickness, integrated with graphitized carbon and performs as outerwear to encapsulate the active particles within the carbon layer. It is seen that the Sn nanoparticles disperse on the surface of Si strains, which would act as transmission sites to contribute to the transmission of electrons and enhance the conductivity of the composite.

In order to evaluate the specific surface area and pore distribution of the composites, BET is implemented and the result is given in Figure 4, and Appendix A. It is proved that the specific surface area of porous Si and Si/Sn@G-C composite are 17.1 and 26.4 m^2^ g^−1^, respectively. The N_2_ isothermal absorption and desorption curves show the type-IV adsorption of both Si/Sn@G-C and porous Si, which indicates the presence of macropores and mesopores. Figure 4b presents the pore size distribution of porous Si and the Si/Sn@G-C composite, verifying the existence of macropores and mesopores. Although the specific surface area of Si/Sn@G-C is larger in comparison to porous Si, the micropores are effectively controlled during the compositing process, which can restrain the irreversible Li^+^ insertion in the micropores. More mesopores and macropores in the composite can promote electrolyte infiltration and is favorable for the transmission of lithium ions, therefore the rate performance of the material is improved [39]. Besides, these pores can relive the huge volume change of Si anode during cycles, resulting in good long-term cycling stability. The BET result of Si@G-C and Si/Sn@G is given in Appendix A, respectively. These composites show the same adsorption type and similar pore size distribution with Si/Sn@G-C, the specific surface area is 25.8 m^2^ g^−1^ and 23.3 m^2^ g^−1^ for Si/Sn@G and Si@G-C, respectively.

The electrochemical performance of the composites and porous Si is fully evaluated and shown in Figure 5, and Appendix A. Figure 5a shows the charging/discharging curves of Si/Sn@G-C composite for the 1st, 20th, 100th and 300th cycles at 1 A g^−1^ within the potential range of 0.01–1.2 V (vs. Li/Li^+^). The composite has an initial discharge capacity of 1425.5 mAh g^−1^ with an initial columbic efficiency (ICE) of 79% and the irreversible specific capacity is 299.9 mAh g^−1^, which is mainly caused by the formation of solid electrolyte interface (SEI) [64]. It is indicated by Figure 5a that the reversible specific capacity remains 1176 mAh g^−1^ after 100 cycles, corresponding to 83% capacity retention relative to the first cycle, and even after 300 cycles, the composite still exhibits obvious electrochemical platform with slight polarization voltage.

Figure 5c reveals the cycle performance of Si/Sn@G-C composite at 500 mA g^−1^ in the potential of 0.01–1.2 V. It is indicated that the composite presents a specific discharge capacity of 1013.3 mA g^−1^ after 100 cycles with high capacity retention (92%). Figure 5d compares the circulation properties of the composites at 1 A g^−1^. The Si/Sn@G-C composite shows the best cycle performance among all the samples, whose specific discharge capacity remains 825.5 mAh g^−1^ after 300 cycles, while Si@G-C and Si/Sn@G remains 588.8 and 559.4 mAh g^−1^, respectively. The best cycle performance of Si/Sn@G-C may result from the structural and kinetic reinforcing role of amorphous carbon and nano-Sn. The rate performance of the prepared composites is displayed in Figure 5b, the charge capacity of Si/Sn@G-C is 1278, 1273, 1197, 1094, 940, and 709 mAh g^−1^ at 0.05, 0.1, 0.2, 0.5, 1, 2 and 5 A g^−1^, respectively. Once the current density returns to 0.2 mA g^−1^, the specific capacity recovers to around 1103 mAh g^−1^, indicating the robustness of Si/Sn@G-C composite. The result reveals that after the addition of Sn and amorphous carbon, the cycle and rate performance of the composite has been significantly improved. That is because Sn can improve the conductivity of the composite, enables the full utilization of the active material and the amorphous carbon can work as bridging agent, prevents the silicon particles from separation by provide effective constrain force [39,44]. We also compare the performance of the Si/Sn@G-C composite with other electrodes based on similar materials (Appendix A), it is obvious that the cycle performance or the ICE of the as prepared sample is not the most prominent, but the sample has good overall performance.

The CV curves of the composites within the voltage of 0.01–1.5 V are depicted in Figure 6a–c. The measurements are performed at a sweep rate of 0.1 mV s^−1^. For the Si/Sn@G-C composite, a clear reduction peak at 0.2 V and oxidation peaks at 0.32 V and 0.48 V can be observed and can be attributed to the silicon alloying/dealloying reactions with Li^+^ (Figure 6a) [16]. The lithiation/delithiation peaks of nano-Sn are unobservable in the CV curves because of the low content and ultrafine particle size of Sn [44]. The intensity of the peaks increases gradually in the following cycles, indicating a gradual activation process of Si-based materials. Figure 6b,c display the CV curves of the Si/Sn@G and Si@G-C composites, which exhibit similar CV peaks with Si/Sn@G-C.

The CV curves of Si/Sn@G-C composite at scanning rates from 0.1 to 0.5 mV s^−1^ are shown in Figure 6d. As the scanning rate increases, the oxidation and reduction current densities increase at the same time. The polarization voltage increases slightly along with the increasing of scanning rate, but obvious reduction and oxidation peaks can be observed, which is because faster scanning rate results in more severe electrode polarization, therefore, increases the irreversibility of active materials [16]. The relationship between the peak current (*i*) and the scan rate (*v*) can be described as *i* = *av^b^*, where *a* and *b* are adjustable parameters. Parameter *b* can be calculated by the slope of the fitted line lg *i* versus lg *v*, and it reflects the main influence factor of the electrochemical reaction, *b* = 0.5 means that electrochemical reaction is controlled by solid-state diffusion, while *b* = 1 indicates that the reaction is dominated by surface reactions [65]. In our work, the lg *i* and lg *v* in the state of oxidation provides a *b* value of 0.67 with a standard error of 0.069 (Figure 6e), showing a diffusion-dominated lithium-storage mechanism.

In order to further testify the positive effect of nano-Sn and amorphous carbon to the electrochemical properties of Si, we performed electrochemical impedance spectroscopy (EIS) measurements (Figure 7). Figure 7a displays the Nyquist plots of the composites before cycles, the semicircle at high frequency region and the straight lines at low frequency region are ascribed to the resistance of charge transfer (R_ct_) and solid-state diffusion of lithium ions, respectively. The value of R_ct_ of the Si/Sn@G-C, Si@G-C and Si/Sn@G anodes before cycling is 170, 266 and 335 Ω, respectively. The reduced electrochemical impedance of Si/Sn@G-C can be attributed to the kinetics improvement by highly conductive nano-Sn and connection enhancement by flexile amorphous carbon [16,44]. The fitted R_ct_ values of the electrodes after 200 cycles is 30, 63 and 50 Ω respectively, which indicates that an activation process occurs during cycles and the Si/Sn@G-C electrode has excellent structural stability during cycle.

To further observe the change of the as prepared electrodes before and after cycles, the SEM are carried out. Figure 8a–c show the SEM images of electrodes before cycles, it is obviously that all the electrodes have a porous morphology. After 200 cycles, the morphology of the Si/Sn@G-C electrode remains intact (Figure 8d), showing the outstanding structure stability. On the contrary, many cracks can be observed on the electrode of Si/Sn@G and Si@G-C (Figure 8e,f), which are possibly attributed to the stress accumulation of Si during the processes of charge/discharge and these cracks will lead to bad electrical contact between active materials and the current collector, resulting in a poor electrochemical performance. This result also demonstrates the reasonable design of Si/Sn@G-C composite toward high-performance LIBs.

## 4. Conclusions

In conclusion, the Si/Sn@G-C composite is synthesized using the micro-sized porous Si, off-the-shelf additive including tin dichloride, PVA and graphite. The morphology property, structure and electrochemical performance of the composite are studied systematically. The pores in micro-sized porous Si can relieve the volume expansion during cycles and provide the transport routes for Li^+^. In addition, the surface-decorated nano-Sn and composited graphite can obviously decrease the electrical resistance of the composite, facilitating the lithiation of the silicon and contributes to the total energy storage capacity of the anode. Furthermore, the amorphous carbon can work as outerwear and bridging agent, prevents the silicon particles from separation by provide effective constrain force and is beneficial for the transmission of electrons, as well as promoting the formation of stable SEI layer. As a result, the Si/Sn@G-C composite shows outstanding cycle and rate performance. It delivers a reversible specific capacity of 1164 mAh g^−1^ at 1 A g^−1^ and maintains 825.5 mAh g^−1^ after 300 cycles, and it also exhibits excellent rate performance of 445 mAh g^−1^ at 5 A g^−1^. This work gives a facile and effective approach to fabricate the Si/Sn@G-C composite with satisfied performance, which will provide reference for the synthesis of next-generation Si-based anode materials.

## Figures and Tables

**Figure 1 materials-14-00920-f001:**
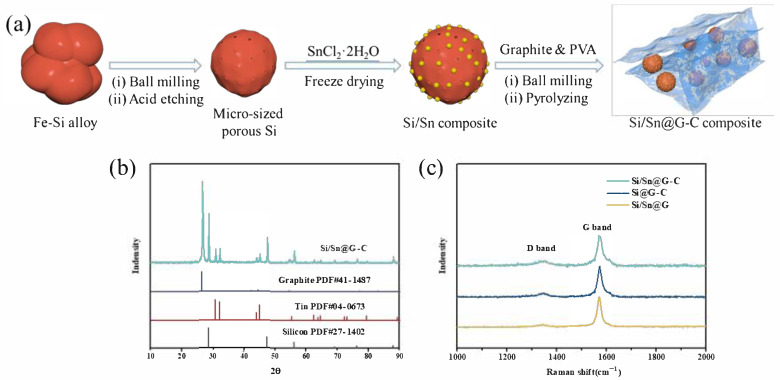
(**a**) Preparation process of Si/Sn@G-C composite; (**b**) XRD pattern of Si/Sn@G-C composite; (**c**) Raman spectrum of the composites.

**Figure 2 materials-14-00920-f002:**
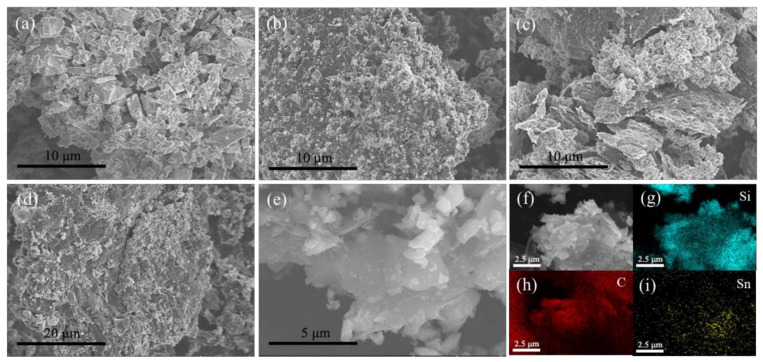
The SEM images of (**a**) micro-sized porous Si, (**b**) Si@G-C composite, (**c**) Si/Sn@G composite, (**d**,**e**) Si/Sn@G-C composite, (**f**–**i**) EDS result of Si/Sn@G-C composite.

**Figure 3 materials-14-00920-f003:**
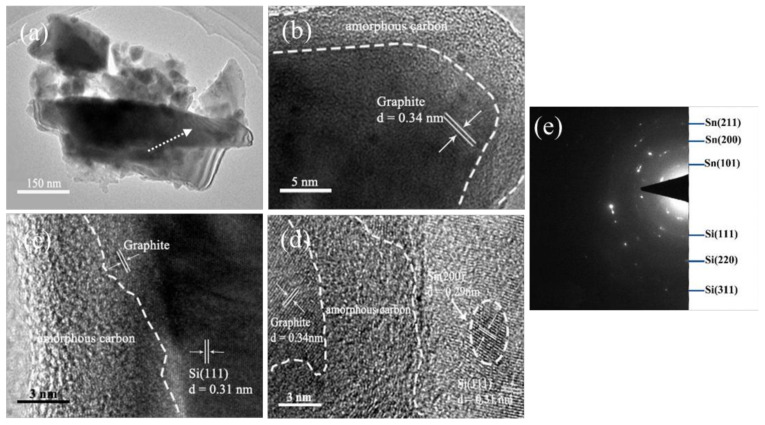
TEM and HRTEM pictures of the Si/Sn@G-C composite. (**a**) TEM image, (**b**–**d**) HRTEM images, (**e**) SAED pattern of the zone arrowed in (**a**).

**Figure 4 materials-14-00920-f004:**
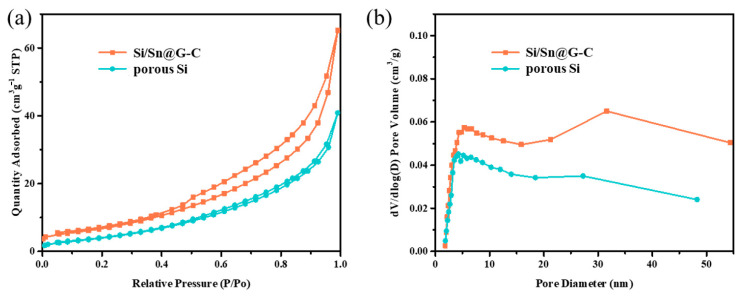
The BET results of the Si/Sn@G-C composite (**a**) the N_2_ adsorption/desorption isotherm and (**b**) pore size distribution.

**Figure 5 materials-14-00920-f005:**
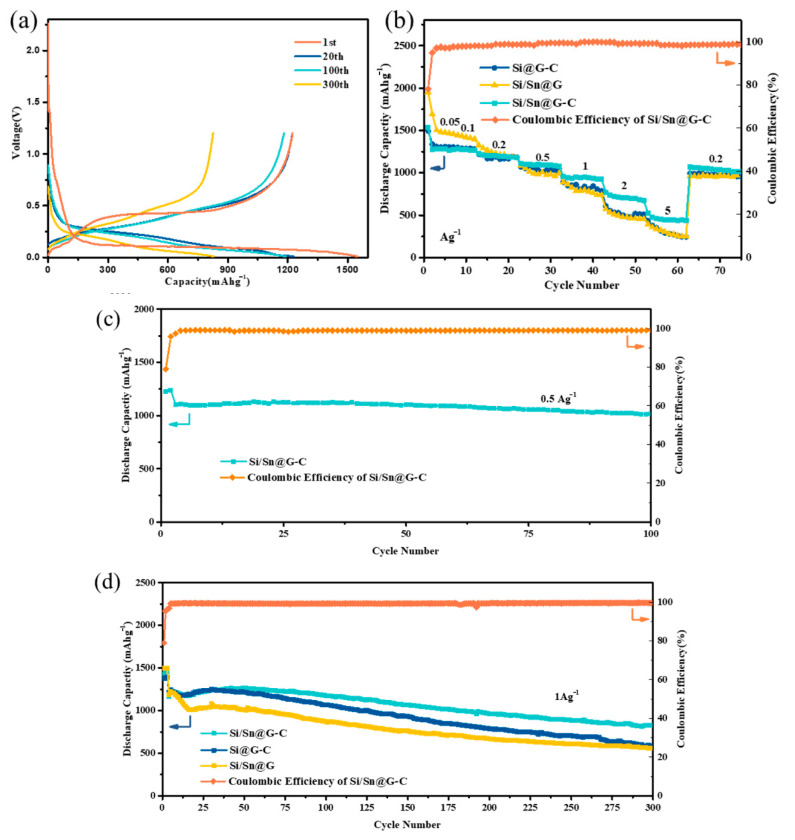
Electrochemical performance of as-prepared samples: (**a**) the charging/discharging voltage profiles of the Si/Sn@G-C anode for the 1st, 20th, 100th and 300th cycles; (**b**) rate performance of the Si/Sn@G-C anode; (**c**) cycle performance of the Si/Sn@G-C anode at 0.5 A g^−1^ in 100 cycles, (**d**) cycling performance of Si/Sn@G-C, Si@G-C and Si/Sn@G anodes at 1 A g^−1^.

**Figure 6 materials-14-00920-f006:**
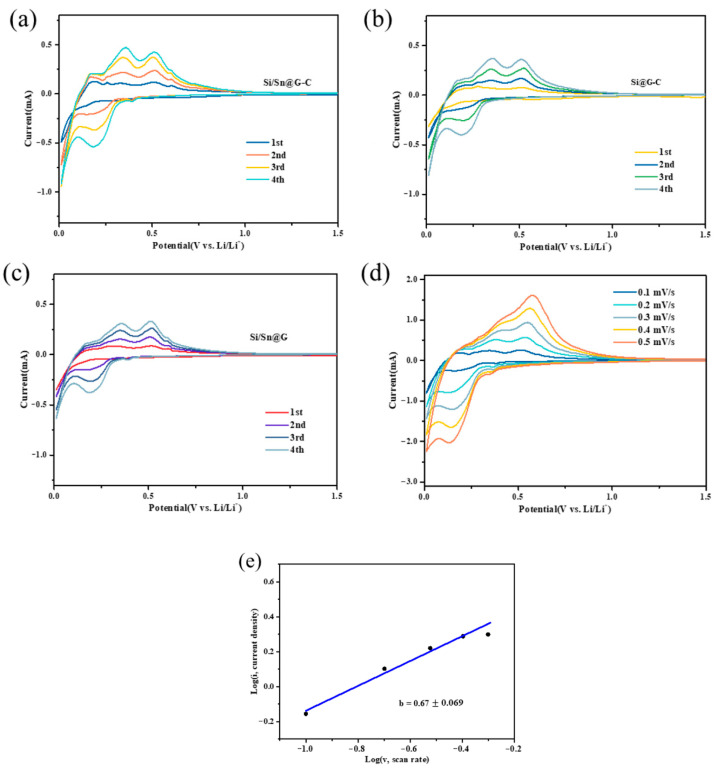
CV curves of the initial four cycles of (**a**) Si/Sn@G-C, (**b**) Si@G-C, (**c**) Si/Sn@G anode at a scan rate of 0.1 mV s^−1^ with a potential range from 0.01–1.5 V vs. Li/Li^+^ and (**d**) CV curves of Si/Sn@G-C composite at different scanning rates; (**e**) log(i)/log(v) plots of Si/Sn@G-C at various scan rates from 0.1 to 0.5 mV s^−1^.

**Figure 7 materials-14-00920-f007:**
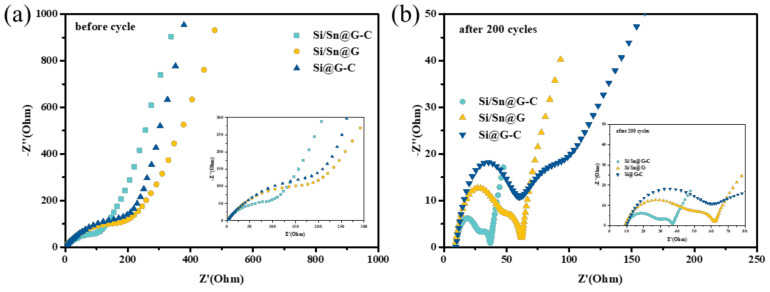
The Nyquist plots of the composites (**a**) before cycling and (**b**) after 100 cycles.

**Figure 8 materials-14-00920-f008:**
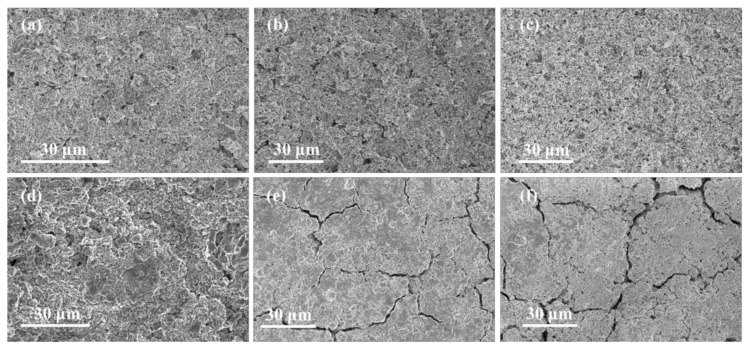
SEM pictures of the electrodes (**a**) Si/Sn@G-C, (**b**) Si@G-C, (**c**) Si/Sn@G before and (**d**) Si/Sn@G-C, (**e**) Si@G-C, (**f**) Si/Sn@G after 200 charge/discharge cycles at 1 A g^−1^.

## Data Availability

The data presented in this study are available on request from the corresponding author.

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
