# Peer review of "Electrochemical Performance Enhancement of Micro-Sized Porous Si by Integrating with Nano-Sn and Carbonaceous Materials"

_materials, 2021, doi:10.3390/ma14040920_

Round 1

Reviewer 1 Report

The authors present the fabrication of Si/Sn@G-11 C composite, improving the initial columbic efficiency and stable reversible specific capacity. The paper is written using good English language, showing little typos. The introduction is good, providing a broad idea of the current literature, although some extra references are required. The “Materials and Method” section is poor, it does not provide enough information, making its reproducibility, by other research groups, very challenging. Moreover, some of the graphs/figures need to be improved/change. 

Consequently, the paper can be considered for publication after these sections have been improved. 

Detailed comments have been added directly to the manuscript. 

Reviewer 2 Report

Title: Electrochemical performance enhancement of micro-sized porous Si by integrating with nano-Sn and carbonaceous materials
Authors: Tiantian Yang, Hangjun Ying, Shunlong Zhang, Jianli Wang, Zhao Zhang and Wei-Qiang Han

This paper was well considered and these experiments has good results using micro-sized porous Si materials. It is worth data for Li-ion battery technology.
Therefore, I can recommend to accept Materials journal.

Author Response

We appreciate the reviewer’s positive comments. Thank you!

Reviewer 3 Report

The manuscript has some merits and could be considered after thorough revision, see attachment.

Reviewer 4 Report

I believe that the paper entitled "Electrochemical performance enhancement of micro-sized porous Si by integrating with nano-Sn and carbonaceous materials" is suitable for publication in Materials after the authors consider the following minor points:

1.You need to include the following references in Introduction along with [4-8]                                                                                              Electrochimica Acta 196 (2016) 294-299

ChemElectroChem 7 (2020) 4289-4302

2. You need to compare your data (capacity and/or stability) with the literature to strengthen the novelty of your work. The role of Sn and Si on the overall electrochemical performance of the carbonaceous materials is clear. But, is this performance better or enhanced compared with other electrodes? This information will be useful to include it.

3. Can you please include a few comments how your synthetic route can provide reference for large-scale synthesis? Could your synthetic route be upscaled? In other words, your synthetic route is a chemical process that contains many steps. How this process can be upscaled? Please include your consideration.

4. Please pay also attention in English grammar. For instance, it is "are corresponded" and not "are correspond".

Reviewer 5 Report

This paper reports on “electrochemical performance enhancement of micro-sized porous Si by integrating with nano-Sn and carbonaceous materials”. The authors presented a facile approach for preparing Si/Sn@G-C composite as anode for LIB. The results reported in this paper could be valuable for publication. However, several points have to be reconsidered for major revision as mentioned below.
1) The authors gave several examples from the literature on using Si@C-G and Si-M alloys as anode materials for LIB in the introduction part but no reports were given about Sn/Si, Sn/Si@G-C and Sn/Si@C. Thus, the literature in the introduction part should be updated including the reports on these related material systems.
2) The authors should also compare the electrochemical performance of their materials with similar materials (e.g., Si@C-G, Sn/Si, Sn/Si@G-C and Sn/Si@C, etc) reported in the literature, I suggest adding a table summarized these results in the electrochemical performance part for a better comparison.
3) The influence of Sn on the electrochemical performance enhancement should be explained in more detail in the electrochemical performance part. Please explain why Sn improves the electrochemical performance of the anode material.
4) Although the authors determined only the amount of C in the anode materials no analysis was performed to quantify the amount of Si and Sn in the anode. The amount of C, Si and Sn are very important to calculate the expected theoretical capacity. Thus, elemental analysis of Si and Sn in the anode materials should be provided. I suggest also the calculation of theoretical capacity based on the obtained values.
5) More details about sample preparations for different characterizations and measurement parameters for each technique should be given in the experimental part. For instance, measurement geometry of the diffractometer, current, and voltage, scan parameters, measurements performed on powder or pellets samples, etc. The same for other techniques.
6) On which part of the sample the diffraction pattern (inset of Fig. 3a) was measured, please highlight it in Fig. 3a. Also, please index the diffraction spots according to the assigned phase.
7) Please explain why the specific surface area of Si/Sn@G-C increases in comparison to porous Si, gas evolution during pyrolysis?
8) The alloying/dealloying of Si and Sn is accompanied by large volume expansion and contraction, which may influence the anode stability. Thus, I suggest performing SEM characterizations on the anode materials after the electrochemical performance test to study the stability of the anode materials.
9) Please define G-C, where it first appears.
10) Please check the manuscript for typos, including grammar and spelling mistakes. For example “Micro-sized, P2, L82”, “release 0the mechanical, P2, L49”

Round 2

Reviewer 3 Report

The manuscript has been improved, the missing experiment/equipment descriptions have been provided. Several points clarified. While the English still needs polishing (like changing the general tense to past, as the experiments were performed some time ago), and fixing sinlular-plural mismatches, etc, the content is much more acceptable for publishing. The wording still requires a lot of attention, some examples:

"cyclic voltammogram is carried out.."

"TGA test, whose results is.."

"spectra are recorded in Figure 1c"

"pores can relive the huge volume change"

The list is far from complete.

Reviewer 5 Report

The authors have carefully revised the manuscript taking in account all the comments of the reviewers. Therefore, I recommended the acceptance of this manuscript in its present form.